

# Enhancing brain tumor diagnosis: an optimized CNN hyperparameter model for improved accuracy and reliability

Abdullah A. Asiri[1], Ahmad Shaf[2], Tariq Ali[2], Muhammad Aamir[2], Muhammad Irfan[3] and Saeed Alqahtani[1]

[1] Radiological Sciences Department, College of Applied Medical Sciences, Najran University, Najran, Najran, Saudi Arabia
[2] Department of Computer Science, COMSATS University Islamabad, Sahiwal, Punjan, Pakistan
[3] Electrical Engineering Department, College of Engineering, Najran University, Najran, Najran, Saudi Arabia

Corresponding author
Ahmad Shaf,
ahmadshaf@cuisahiwal.edu.pk

## ABSTRACT

Hyperparameter tuning plays a pivotal role in the accuracy and reliability of convolutional neural network (CNN) models used in brain tumor diagnosis. These hyperparameters exert control over various aspects of the neural network, encompassing feature extraction, spatial resolution, non-linear mapping, convergence speed, and model complexity. We propose a meticulously refined CNN hyperparameter model designed to optimize critical parameters, including filter number and size, stride padding, pooling techniques, activation functions, learning rate, batch size, and the number of layers. Our approach leverages two publicly available brain tumor MRI datasets for research purposes. The first dataset comprises a total of 7,023 human brain images, categorized into four classes: glioma, meningioma, no tumor, and pituitary. The second dataset contains 253 images classified as "yes" and "no." Our approach delivers exceptional results, demonstrating an average 94.25% precision, recall, and F1-score with 96% accuracy for dataset 1, while an average 87.5% precision, recall, and F1-score, with accuracy of 88% for dataset 2. To affirm the robustness of our findings, we perform a comprehensive comparison with existing techniques, revealing that our method consistently outperforms these approaches. By systematically fine-tuning these critical hyperparameters, our model not only enhances its performance but also bolsters its generalization capabilities. This optimized CNN model provides medical experts with a more precise and efficient tool for supporting their decision-making processes in brain tumor diagnosis.

## INTRODUCTION

Brain tumors, the leading cause of demise with the lowest survival rate among cancers, pose challenges in early detection due to their asymmetrical shapes and dispersed borders. Accurate analysis at the initial stage is crucial for precise medical interventions and saving lives. Brain tumors manifest as benign (non-cancerous) or malignant (cancerous) types,

with primary and secondary distinctions based on origin (*Siegel, Miller & Jemal, 2015*; *Sauer, 2019*).

Common types include meningioma, glioma, and pituitary cancer. Meningiomas originate from the meninges, gliomas from glial cells supporting nerve function, and pituitary tumors impact various bodily processes (*Abiwinanda et al., 2019*; *Abir, Siraji & Khulna, 2018*). Understanding these types and their characteristics is vital for effective diagnosis and treatment, supporting healthcare professionals in providing appropriate care.

Identifying and estimating the duration of brain tumors presents a significant challenge in medical diagnostics. The datasets used for this purpose comprise images obtained through various diagnostic techniques, including biopsies, spinal taps, computed tomography scans, and magnetic resonance imaging. These datasets undergo segmentation, classification, and feature extraction based on specific requirements. Deep learning techniques have emerged as highly effective tools in this domain, particularly in brain tumor detection. Unlike traditional methods focusing on segmenting tumors for classification and feature extraction, deep learning approaches employ classification algorithms to identify and categorize brain tumors. Deep learning, a branch of machine learning and artificial intelligence, mimics how humans acquire knowledge. Deep learning algorithms handle complex and abstract tasks, surpassing the performance of traditional linear machine learning systems that are more suited for smaller datasets. By harnessing the power of deep learning, accurate and efficient brain tumor diagnosis becomes a reality, potentially revolutionizing the field of medical imaging and enhancing patient care (*Naseer et al., 2020*).

Accurate segmentation of brain tumors is crucial for cancer diagnosis, treatment planning, and outcome evaluation. However, manual segmentation is a laborious, time-consuming, and challenging task. To overcome these limitations, researchers have extensively investigated automatic and semi-automatic brain tumor segmentation methods (*Núñez-Martín, Cervera & Pulla, 2017*). A generative or discriminative model is the foundation for automatic and semi-automatic segmentation. These methods are built upon either generative or discriminative models. The discriminative model relies on image features to categorize normal and malignant tissues, while the generative model utilizes probabilistic information obtained from images for brain tumor segmentation. Classification techniques, such as support vector mechanism (SVM) and random forest, are commonly employed in discriminative models (*Kleesiek, 2014*) based on visual features like local histograms, image textures, and structure tensor eigenvalues. These research efforts aim to develop efficient and reliable segmentation approaches that can alleviate the burden of manual segmentation, enabling accurate tumor delineation and facilitating treatment planning and evaluation (*Meier et al., 2014*).

Deep learning algorithms are now often used for object identification, classification, and feature extraction. Mainly, convolutional neural networks are acknowledged as an outstanding method for semantic picture segmentation, and convolutional neural networks-based algorithms performed and generated reliable results (*Long, Shelhamer & Darrell, 2015*). The most advanced mechanism, convolutional neural network (CNN), can

learn through the representation of data and can predict and draw conclusions depending on available data. It successfully accomplished picture categorization and feature extraction tasks by extracting low and high-level information through self-learning. Although a large training dataset was necessary, CNN-based approaches effectively formulate predictions and conclusions. The implementation of CNN is problematic in this situation since brain tumor is a clinical research topic, and the dataset is constrained.

Deep learning, including CNNs, can be used effectively with smaller datasets, it relies on a transfer learning strategy built upon two hypotheses: (1) fine-tuning the ConvNet, and (2) freezing the ConvNet layers. Transfer learning techniques involve using two datasets: a large dataset known as the base dataset and a smaller dataset used for training purposes. A pre-trained network is initially applied to the large dataset, extracting valuable information. This extracted information is transferred and utilized as input for the smaller dataset (*Rehman et al., 2019*). This process, known as fine-tuning, enables the adaptation of the pre-trained network to the specific characteristics of the smaller dataset. By adopting transfer learning, the information acquired from the base dataset can be effectively fine-tuned, enhancing the performance of the CNN on the target task using the smaller dataset.

Our study focuses on the crucial aspect of hyperparameter tuning in CNN models for brain tumor diagnosis. While our work specifically focuses on fine-tuning, its contribution lies in the effective optimization of hyperparameters, which is a critical step in achieving accurate and reliable results in medical image analysis. Our main contributions are as follows:

1. Optimized hyperparameter tuning: We propose a fine-tuned CNN hyperparametric model that systematically optimizes key hyperparameters, including the number and size of filters, stride padding, pooling techniques, activation functions, learning rate, batch size, and number of layers. This optimization process is designed to enhance the model's performance and improve its generalization capabilities.
2. Improved diagnostic precision: Our fine-tuned CNN model showcases impressive results in terms of various performance metrics, including average precision, recall, F1 score, and accuracy. By achieving high accuracy rates (*e.g.*, 96% accuracy for dataset 1), we provide a more precise and efficient tool for medical experts to aid in brain tumor diagnosis.
3. Comparative analysis: We perform a comprehensive comparison of our fine-tuned approach with existing techniques. The comparison clearly demonstrates that our method outperforms these existing methods, reinforcing the effectiveness of our hyperparameter optimization strategy.

The remaining sections of the manuscript are structured as follows: related work, which describes the current advancement and their limitations; methodology, the specifics of our hyperparameter-based CNN model utilized with two distinct brain tumor datasets, discussing their characteristics and preprocessing steps; results, which designates the outcomes of the applied model; and conclusion, which provides a summary of the article and future directions

## RELATED WORK

*Deng et al. (2009)* implemented CNNs using a sizable dataset called ImageNet and successfully obtained the best result on visual recognition tests. Its limitations include biases, label noise, class imbalance, fixed resolutions, and potential domain shift challenges The best outcomes were obtained when CNNs were applied to image classification and detection datasets by *Everingham et al. (2015)*. Its limitations include limited diversity, fixed object categories, imbalanced classes, static scenes, annotations' limitations, static object views, and potential lack of semantic context and temporal variation In a study (*Cheng et al., 2015*), the Figshare dataset explored an alternative algorithm for enhancing tumor regions as areas of interest, which were subsequently divided into sub-sections. The approach involved extracting features such as intensity histogram, gray-level co-occurrence matrix, and employing a bag of words (BoW) model. Using a ring-form partitioning technique, the algorithm achieved impressive accuracy values of 87.54%, 89.72%, and 91.28%.

Meningioma, glioma, and pituitary tumors are all classified by *Ismael & Abdel-Qader (2018)* with a 91% accuracy rate. Using a 2D Gabor filter and MRI, statistical characteristics were retrieved on MRI brain tumor dataset. Multilayer perceptron neural networks were trained using back-propagation for classification purposes. *Shakeel et al. (2019)* applied fractional and multi-fractional dimension algorithms for a feature and essential feature extraction on MRI brain tumor dataset. A classification approach was suggested, and machine learning with back-propagation improved the performance of brain tumor detection. However, the generalizability of the results to different datasets or tumor types may be limited, and the absence of comparative analysis with other state-of-the-art methods raises questions about the relative effectiveness of their proposed approach.

*Xie et al. (2022)* presented a comprehensive review of CNN techniques applied to brain tumor classification from 2015 to 2022. It highlights the advancements, challenges, and achievements in this domain, providing insights into state-of-the-art methodologies. The article concludes with a discussion of future perspectives and potential directions for further research in brain tumor classification using CNNs. By using five alternative CNN designs, *Abiwinanda*'s *(2018)* experiment obtained 98.51% accuracy on training sets of brain tumor datasets consisting of 3064 T-1 weighted CE-MRI images publicly available *via* Figshare. Providing insights into the choice of hyperparameters, training strategies, and data augmentation techniques would enhance the reproducibility and understanding of the proposed classification method. In order to distinguish between benign and malignant tissues in MRIs brain tumor dataset, *Al-Ayyoub et al. (2012)* used MATLAB and ImageJ. Nearly ten different features were extracted from the MRIs to identify brain tumors. Potential limitations of the study may include the need for a more detailed description of the machine learning algorithms and methodologies employed, such as the specific features extracted from the images and the choice of classifiers used. *Parihar (2017)* suggested a CNN-based approach that entails intensity normalization during pre-processing on MRIs dataset, CNN architecture for classification, and tumor

classification during post-processing. The study might include the need for more extensive experimentation and validation on diverse datasets to establish the robustness and generalizability of the CNN-based segmentation approach.

In order to classify tumors into meningioma, glioma, and pituitary tumors, *Sultan, Salem & Al-Atabany (2019)* used two publically accessible datasets known as T1-weighted contrast-enhanced images and Cancer Imaging Archive (TCIA) as well as two deep-learning models—a second model graded gliomas as Grade II, III, or IV. The need for comprehensive evaluation on diverse datasets to establish the generalizability of the proposed deep neural network across various types of brain tumors and imaging conditions. Using a relatively small dataset from CE-MRI, *Ismael, Mohammed & Hefny (2020)* experimented to determine the prevalence of three different types of tumors, including meningioma, gliomas, and pituitary tumors, and they found rates of 45%, 15%, and 15%, respectively. The study's reliance on ResNet architecture might require careful consideration of model complexity and potential overfitting, especially with limited data.

A frequent type of brain tumor is called a glioma, which is further divided into high-grade and low-grade gliomas. The severity of the tumor is taken into account when assigning these grades. Both have different classifications, benign and cancerous, respectively. The research study in *Vinoth & Venkatesh (2018)* suggested a CNN technique to identify low and high-grade tumors on the MRI brain tumor dataset. An effective SVM classifier categorizes benign and malignant tumors based on the constraints and outcomes collected. However, potential limitations of the study might include the need for careful parameter tuning and validation of the CNN and SVM models to ensure optimal performance. A work by *Rehman et al. (2020)* uses CNN architecture and transfer learning to categorize brain tumors. Three deep CNN architectures AlexNet, GoogLeNet, and VGGNet were applied to the target dataset's MRIs to control the type of tumor. The framework's performance could be influenced by factors such as the availability and quality of labeled data, and the generalizability of the approach to various tumor subtypes and imaging modalities might require further validation. To classify images of brain tumors, the author in *Swati et al. (2019)* proposed a block-wise fine-tuning technique using transfer learning and fine-tuning on the T1-weighted contrast-enhanced magnetic resonance images (CE-MRI) benchmark dataset. The results with traditional machine learning and deep learning CNN approaches were compared; under five-fold cross-validation, the applied method had an accuracy of 94.82%. It could include the potential sensitivity of the transfer learning approach to variations in dataset characteristics, potentially leading to suboptimal performance when applied to datasets with significantly different imaging conditions or tumor characteristics. The latest literature comparison of different techniques is also given in Table 1.

## METHODOLOGY

This section explains the methodology's overall structure. Here is a detailed explanation of every parameter used in the proposed system. Graphical representation of the proposed work has been illustrated in Fig. 1.

**Table 1 Literature comparison of existing work.**

| Dataset | Author | Methodology | Accuracy (%) | Limitations |
|---------|--------|-------------|--------------|-------------|
| MRI | Mahmud, Mamun & Abdelgawad (2023) | CNN | 93.3 | More work can be performed to correctly identify brain cancers by using individual patient information gathered from any source. |
| MRI | Alsaif et al. (2022) | VGG19 | 93 | More datasets, complex structure, and augmented techniques may be used to enhance the model accuracy. |
| BraTS18 | Rehman et al. (2021) | CNN | 92.67 | More techniques can be employed to enhance the accuracy results on BraTS dataset. |
| BraTS15,17 | Nema et al. (2020) | GAN-Net | 94.01 | Limited dataset diversity and size may impact generalization to different brain tumor types. Lack of detailed explanation of data preprocessing steps and augmentation techniques. |
| MRI | Deng et al. (2019) | FCNN | 90.89 | FCNN may struggle with capturing complex spatial patterns in brain tumor images, leading to limited accuracy. |
| BraTS18 | Sun et al. (2019b) | CA-CNN | 61.0 | The study's limitations include the use of a relatively small dataset and the lack of ground truth data. |
| BraTS | Gonella (2019) | V-Net | 85 | The study only used a single dataset, the BraTS 2018 dataset. It is important to evaluate the model on other datasets to ensure that it generalizes well to new data. |
| BraTS18 | Kuzina, Egorov & Burnaev (2019) | U-Net-RI, U-Net-PR | 74 | The study did not compare the proposed model to other methods that use transfer learning. The study did not evaluate the computational complexity of the model. |
| BraTS17,2015 | Chen, Ding & Liu (2019) | U-Net, DeepMedic, MLP | 89 | Limited discussion of the model's sensitivity to hyperparameters and optimization choices. |
| BraTS17 | Mlynarski et al. (2019) | 2D-3D Model, U-NET | 91.8 | The study did not compare the proposed model to other methods that use long-range context. The study did not explore the use of different CNN architectures for tumor segmentation. |
| MRI | Thaha et al. (2019) | CNN, ECNN | 92 | The study did not use ground truth data to evaluate the performance of the model. This makes it difficult to compare the results to other studies that use ground truth data. |
| BraTS13,15 | Havaei et al. (2017) | CNN | 88 | It does not provide sufficient details about how this architecture is different from traditional CNNs or how it contributes to solving the problem of brain tumor segmentation. |
| BraTS13, 2015 | Pereira et al. (2016) | CNN | 78 | The relatively lower accuracy of 78% suggests potential limitations in feature extraction or model complexity. |
| MRI | Zhai & Li (2019) | VGG-16 | 82.2 | VGG-16 may struggle with balancing model complexity and overfitting in brain tumor classification tasks. |
| BraTS15 | Sun et al. (2019a) | 3D CNN | 84 | Computational complexity and memory requirements of 3D CNN may limit its applicability on larger datasets. The study did not evaluate the performance of the model on different imaging modalities. |
| MRI | Pereira et al. (2018) | RBM-RF | 84 | Limited discussion on the choice and hyperparameters of the RBM-RF model components. |
| BraTS16 | Hoseini, Shahbahrami & Bayat (2018) | DCNN | 90 | Limited exploration of alternative architectures and their impact on accuracy. |
| BraTS 13,15,16 | Zhao et al. (2018) | FCNNs, CRF-RNN | 84 | CRF-RNN may face challenges in capturing long-range dependencies and spatial relationships in complex tumor images. |
| BraTS17 | Saouli, Akil & Kachouri (2018) | XCNet-ELOBA$_\lambda$ | 89 | Limited insight into how the proposed XCNet-ELOBA$_\lambda$ architecture compares to other state-of-the-art methods. |

| Table 1 (continued) | | | | |
|---|---|---|---|---|
| Dataset | Author | Methodology | Accuracy (%) | Limitations |
| MRI | *Charron et al. (2018)* | 3D-CNN, DeepMedic | 79 | 3D-CNN may face challenges in efficiently processing 3D volumes, potentially impacting performance. |
| BraTS15 | *Liu (2018)* | DCR, ResNet-50 | 87 | Limited discussion on the challenges and implications of using deep residual networks (ResNet) for tumor classification. |
| BraTS15 | *Iqbal et al. (2018)* | SkipNet, SENet, IntNet (VGG) | 90 | The choice of architectures may not fully exploit the unique features of brain tumor images. |
| BraTS15 | *Cui et al. (2018)* | TLN/ITCN, FCN | 89 | Limited analysis of the trade-offs between the proposed architectures and their impact on accuracy. |
| MRI | *Wang et al. (2018)* | CNN | 86.1 | Possible challenges in effectively capturing subtle and intricate tumor patterns using a traditional CNN. |
| MRI | *Wang et al. (2019)* | WRN, ResNet | 91 | Limited discussion on the potential limitations of wide residual networks (WRN) in tumor classification tasks. |
| BraTS17 | *Zhang (2017)* | CNN | 72 | Lower accuracy of 72% suggests that the chosen CNN architecture may not effectively capture relevant features. |
| IBSR2 | *Gottapu & Dagli (2018)* | DenseNet, Growth rate, Bottleneck | 92 | Limited analysis of how the growth rate and bottleneck architecture impact accuracy and training dynamics. |

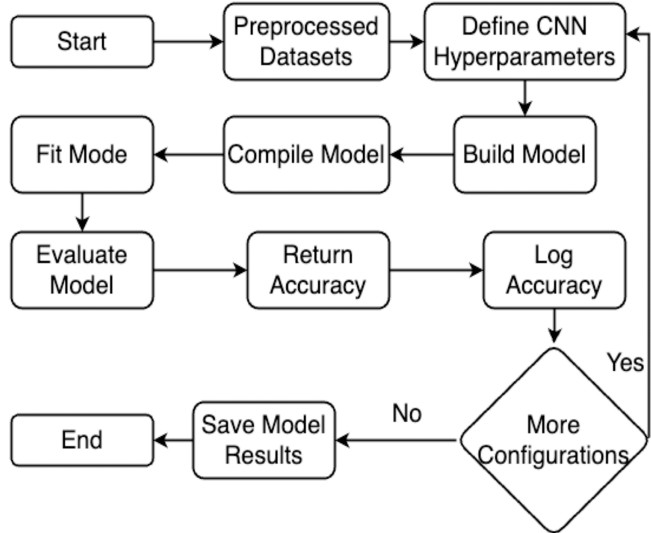

**Figure 1  Flowchart of the proposed work.**

## Dataset details

This study utilized the two brain tumor MRI datasets publicly available at Kaggle. The dataset 1 comprises a total of 7,023 images of the human brain having dimensions of 512 × 512 and JPG format. It consists of four classes: glioma (1,621), meningioma (1,645), no tumor (2,000), and pituitary (1,757). The "no tumor" class images were sourced from the Br35H dataset. Figure 2 illustrates the different categories present in the dataset, including no tumors, meningioma, pituitary, and glioma. The dataset 2 consists of 253 images with

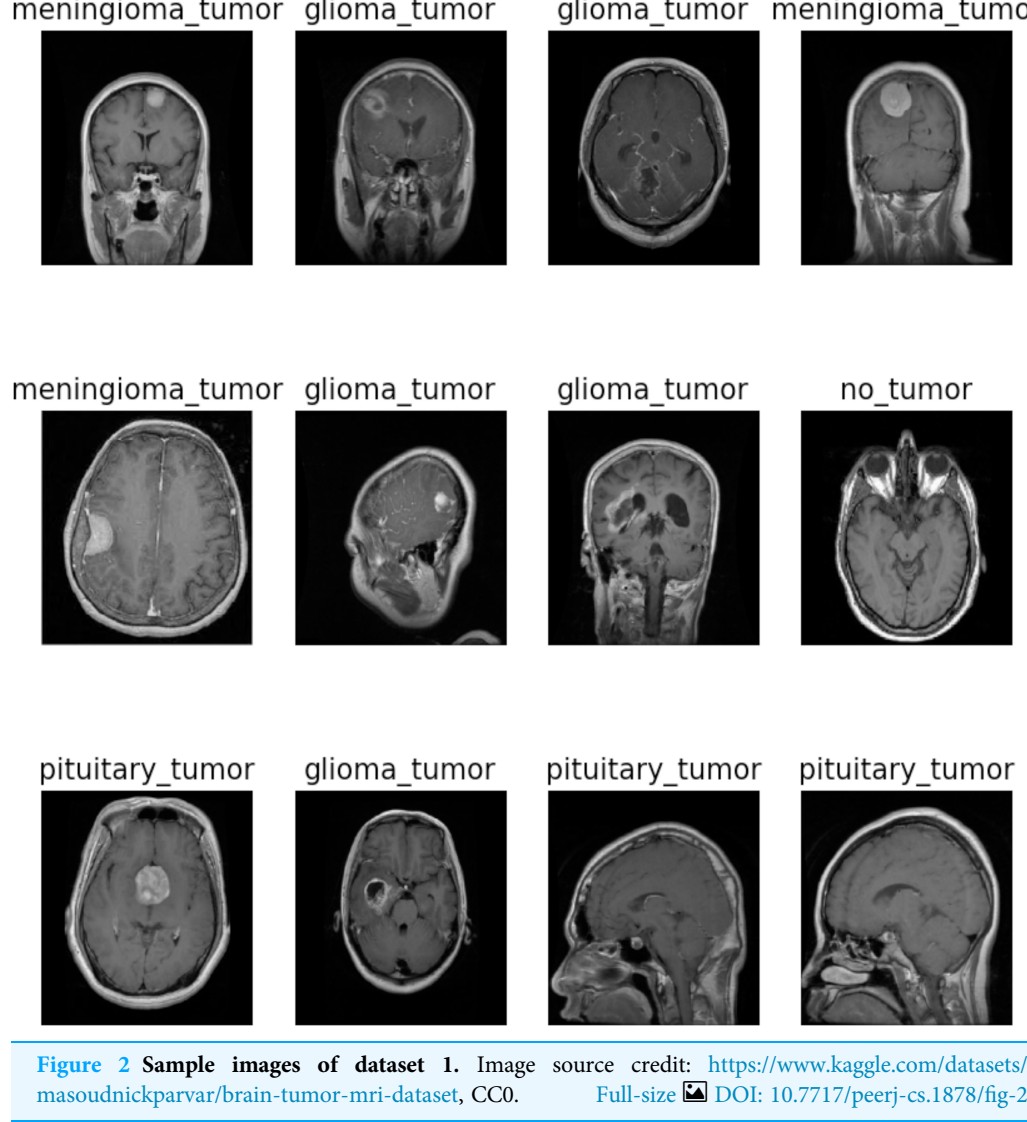

**Figure 2 Sample images of dataset 1.** Image source credit: https://www.kaggle.com/datasets/masoudnickparvar/brain-tumor-mri-dataset, CC0.

yes (155) and no (98) classes. To train and evaluate the hyperparameter-tuned model for brain tumor detection, the datasets were divided into training, validation, and testing sets for dataset 1 and 80:20 ratio of training and testing for dataset 2. A detailed description of the dataset can be found in Tables 2 and 3.

## Pre-processing

Pre-processing a brain tumor MRI dataset involves several steps to optimize the data for analysis and modeling. This includes addressing challenges like varying resolutions and intensity ranges in MRI images. Rescaling the images to a standardized resolution ensures consistency across the dataset while normalizing intensity values enhances subsequent algorithms and models. Techniques like rescaling pixel values to a specific range (0, 1) or using z-score normalization are commonly employed. Aligning the MRI images to a standard reference frame is essential due to variations in position and orientation. Image

**Table 2 Dataset 1 details and distribution.**

**Dataset 1**

| Class name | Num. of images | Training | Testing | Validation |
|---|---|---|---|---|
| Glioma | 1,621 | 1,021 | 300 | 300 |
| Meningioma | 1,645 | 1,033 | 306 | 306 |
| No tumor | 2,000 | 1,190 | 405 | 405 |
| Pituitary | 1,757 | 1,157 | 300 | 300 |
| Total | 7,023 | 4,401 | 1,311 | 1,311 |

**Table 3 Dataset 2 details and distribution.**

**Dataset 2**

| Class name | Num. of images | Training | Testing |
|---|---|---|---|
| Yes | 155 | 124 | 31 |
| No | 98 | 78 | 20 |
| Total | 253 | 202 | 51 |

registration techniques facilitate spatial alignment, enabling improved comparison and analysis. Additionally, addressing noise and artifacts in MRI images caused by factors like patient motion or equipment variations is crucial. Denoising techniques like Gaussian smoothing or non-local means filtering effectively reduce noise while preserving vital details. These pre-processing steps optimize the brain tumor MRI dataset, enabling accurate analysis and meaningful insights.

## Hyperparameters of convolutional neural networks for training

Hyperparameters in CNNs are parameters not learned during the training process but instead set by the user prior to training. The input layer is the parameter where the CNN training process starts, and the classification layer is where it ends the process in a feed-forward fashion. Nevertheless, the opposite process begins with classification and moves through the first convo layer. Neuron J sends information in a forward fashion computed according to Eq. (1) to the value of neurons N in layer L. The non-linear ReLU function determines the output, as indicated in Eq. (2).

$$IP_N^L = \sum_{J=1}^{N} W_N J^L x_j + b_N \tag{1}$$

$$OP_N^L = \max(0, IP_N^L) \tag{2}$$

where *IP*: input, *OP*: output, *W*: weight, and *b*: neuron number. All neurons use Eqs. (1) and (2) to construct output results and form the non-linear activation function by taking input values. The pooling layer employs a k × k window to gather the results to calculate maximum average feature values. Equation (3) explains how to perform this calculation using the SoftMax function for each tumor type.

$$OP_N^L = \frac{e^{IP_N^L}}{\sum_N e^{OP_k^L}} \qquad (3)$$

The back-propagation cost function is calculated by minimizing the new weights, as presented in Eq. (4).

$$C = -\frac{1}{S}\sum_N^S IP(X(a^i/b^i)) \qquad (4)$$

In the training process, the letter S represents the training set sample, while $b^i$ represents the ith sample of the training set with its corresponding label $a^i$. The probability of classification, denoted as $X(a^i/b^i)$, is used to minimize the cost C through the stochastic gradient function. To calculate the weights of each convolutional layer $L$, the weight of the convolutional layer $L$ at iteration t is represented by $W_L^t$, as depicted in Eq. (5).

$$W_L^{t+1} = W_L^t + V_L^{t+1} \qquad (5)$$

Here $W_L^t$: weight of the convolutional layer $L$, $V_L^{t+1}$ is the updated weight value at iteration $t$. The essential component of convolutional neural networks, feature extraction, is made possible by the convolutional layer. Several filters to extract features are present in this layer. The resultant value and layer sizes are evaluated by using Eqs. (6) and (7); correspondingly, here $n_L^i$ is the resultant feature map of the images, $\sigma$ is the function of activation, $y_L$ is the input width and $x_L^i \in f, z^i \in f$ is the filter ($f$) channels.

$$n_L^i = \sigma(x_L^i - y_L + z^i) \qquad (6)$$

$$Convolutional\ layer\ output\ size = \left(\frac{input\ size - filter\ size}{stride}\right) + 1 \qquad (7)$$

A convolutional neural network often employs the pooling layer after each convolutional layer. This layer manages the parameters, which are also in charge of overfitting. The most widely used pooling layer is max pooling, which performs distinct functions from other layers like min pooling and average pooling. Equations (8) and (9) are used to determine the output and size of the pooling layer when x is the output and R is the pooling region, respectively.

$$Plo_{i,j} = \max_{r,s\in R} \qquad (8)$$

$$Pooling\ layer\ output\ size = \left(\frac{convolutional\ layer\ output\ size - Pooling\ Size}{stride}\right) + 1 \qquad (9)$$

## Fine-tuning of CNN hyperparameters

As depicted in Algorithms 1 and 2, optimizing various aspects of the network's architecture and the training procedure is necessary to fine-tune the hyperparameters of a CNN for brain tumor detection, classification. The quantity and size of filters control

convolutional layer depth and receptive field. Consider alternative numbers, such as (16, 32, 64), and different filter sizes, like [3 × 3, 5 × 5]—altering the stride value to regulate the output feature maps' spatial dimensions. The spatial resolution is decreased with a more excellent stride, whereas the spatial resolution is increased with a smaller stride. The input volume's spatial size is preserved during convolutional processes by choosing the suitable padding parameters. Padding may reduce the impact of margins without retaining crucial spatial information.

The varied pooling methods used during the tests, such as min, max, or average pooling, aid in capturing invariant information and lower the spatial dimensions. ReLU, sigmoid, and tanh are three appropriate activation functions for the CNN layers that introduce non-linearity and allow the network to model complicated relationships in the input. A lower learning rate may necessitate more repetitions but can produce better convergence. In contrast, a more significant learning rate may result in faster convergence but runs the risk of overshooting the ideal solution. The experiment with various batch sizes will decide how many training examples will be processed in each iteration. Different network depths can be achieved by changing the number of convolutional and fully connected layers. Regularization techniques like dropout or L2 regularization improve generalization and reduce overfitting. Various optimization techniques, which regulate how the network's weights are changed during training, are being tested, including stochastic gradient descent (SGD) and Adam.

## Working of hyperparameter CNN

Hyperparameteric CNN refers to a CNN model that utilizes hyperparameters to configure its architecture and optimize its performance. The grid search hyperparameters create a grid of possible values for each hyperparameter that we want to tune including the number of filters (num_filters), number of units (num_units), dropout rate (dropout), and optimizer. These hyperparameters are crucial in determining the model capacity, regularization, and learning behavior. As shown in Figs. 3 and 4 for Dataset 1 and Dataset 2, the given data depicts that the different combinations of hyperparameter values are tested, and their corresponding accuracies are recorded, tested, and their corresponding accuracies are recorded. The hyperparameter values are varied for num_filters, num_units, dropout, and optimizer, while the accuracy represents the model's performance with those specific hyperparameter settings.

### *Comparison of hyperparameter values and accuracy*

The various combinations of hyper parameters values and their corresponding accuracies are given in Fig. 2 for dataset 1 and dataset 2. The given data show the following comparison of various accuracies for different parameters.

- **num_filters:** The number of filters determines the number of feature maps extracted by the convolutional layers. Higher values of num_filters may allow the model to capture more complex patterns but can also increase computational requirements.

**Algorithm 1** Algorithm for hyperparameter tuning of CNN.

1: Initialize hyperparameters: $\alpha = [16, 32, 64]$ $\beta = [64, 128, 256]$ $\gamma = [0.1, 0.2]$ $\delta = ['adam', 'sgd']$

2: Set the evaluation metric: METRIC_ACCURACY = 'accuracy'

3: Create a file writer for logging: Create a file writer for 'logs/hparam_tuning'

4: Configure hyperparameter tuning: Set `hparams_config` with hyperparameters $[\alpha, \beta, \gamma, \delta]$ and metric [METRIC_ACCURACY]

5: **Function** train_test_modelhparams

6: Create a sequential model

7: Add Conv2D and MaxPool2D layers using $\alpha$ and `activation=tf.nn.relu`

8: Add another Conv2D and MaxPool2D layers using $\alpha$ and `activation=tf.nn.relu`

9: Flatten the output

10: Add a dense layer with $\beta$ and `activation=tf.nn.relu`

11: Add a dropout layer with $\gamma$

12: Add a dense output layer with 4 classes and `activation=tf.nn.softmax`

13: Compile the model using $\delta$, `loss='sparse_categorical_crossentropy'`, and `metrics=['accuracy']`

14: Train the model on $x\_train$ and $y\_train$ for a specified number of epochs

15: Evaluate the model on $x\_test$ and $y\_test$ and get the accuracy

16: **Return** accuracy **EndFunction**

17: **Function** run run_dir, hparams

18: Create a file writer for the run directory

19: Set the hyperparameters using `hparams`

20: Train and evaluate the model using `train_test_model` function

21: Write the accuracy metric to the summary file

22: **EndFunction**

23: Set `session_num` to 0

24: **for** $\alpha\_value$ in $\alpha$ **do**

25:   **for** $\beta\_value$ in $\beta$ **do**

26:     **for** $\gamma\_value$ in $\gamma$ **do**

27:       **for** $\delta\_value$ in $\delta$ **do**

28:         Create a dictionary `hparams` with current hyperparameter values

29:         Set `run_name` as "run-" + `session_num`

30:         Print '—Starting trial:' + `run_name`

31:         Print the hyperparameters {h.name: hparams [h] for h in hparams}

32:         Call `run` with run_dir as 'logs/hparam_tuning/' + `run_name` and `hparams`

33:         Increment `session_num` by 1

34:       **end for**

35:     **end for**

36:   **end for**

37: **end for**

| Algorithm 1 (continued) |
|---|
| 38: Select best hyperparameters: |
| 39: Let `best_accuracy` = 0 |
| 40: Let `best_hyperparameters` = {} |
| 41: **for** each summary file in 'logs/hparam_tuning' **do** |
| 42:    Read and store accuracy metric for each run |
| 43:    **if** accuracy is greater than `best_accuracy` **then** |
| 44:       Update `best_accuracy` to current accuracy |
| 45:       Update `best_hyperparameters` with hyperparameters of current run |
| 46:    **end if** |
| 47: **end for** |

| Algorithm 2 **Fine-tuned hyperparameter tuning of CNN.** |
|---|
| 1: Train final model with best hyperparameters: |
| 2: Create new instance of model using `best_hyperparameters` |
| 3: Train model on entire training dataset for specified number of epochs |
| 4: Evaluate model on testing dataset and get final accuracy |
| 5: Print best hyperparameters and final accuracy achieved |

- **num_units:** The number of units represents the size of the fully connected layers. It controls the model's capacity to learn complex relationships in the data. The values used range from 64 to 256. Higher values of num_units generally allow the model to capture more intricate patterns, but they can also increase the risk of overfitting if not balanced appropriately.
- **Dropout:** Dropout is a regularization technique that helps prevent overfitting by randomly dropping out a fraction of the units during training. A dropout rate of 0.1 or 0.2 is used in the experiments. Higher dropout rates provide more regularization but can potentially decrease the model's learning capacity.
- **Optimizer:** The optimizer determines the algorithm used to update the model weights during training. Two optimizers are used: Stochastic Gradient Descent (SGD) and Adaptive Moment Estimation (Adam). Adam is an adaptive optimizer that dynamically adjusts the learning rate, while SGD uses a fixed one. Adam generally performs well in a wide range of scenarios, but the choice between the two can depend on the specific problem and dataset.
- **Accuracy:** Accuracy represents the model's performance on the validation or test set. It indicates the proportion of correctly classified samples. The accuracy values range from 0.75 to 0.96, with different hyperparameter settings achieving varying levels of accuracy. It is important to note that accuracy alone does not provide a

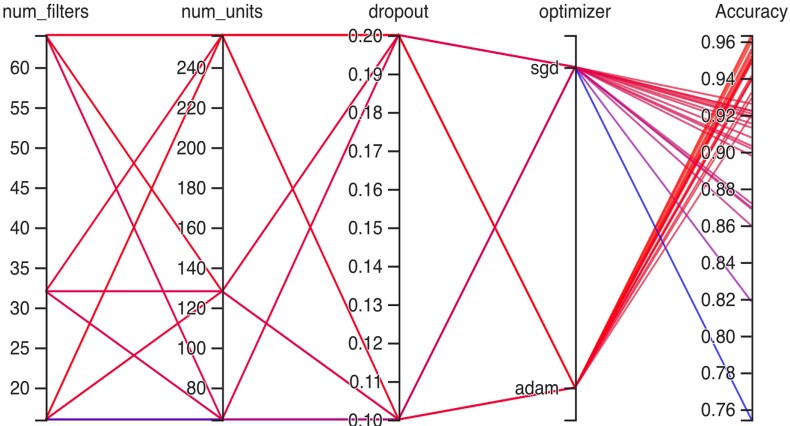

**Figure 3 Working architecture of fine-tuned hyperparametric CNN model for dataset 1.**

comprehensive evaluation of the model performance, and other evaluation metrics should be considered, such as precision, recall, and area under the curve (AUC)-receiver operating characteristic curve (ROC).

In Fig. 3 the x-axis has the following parameters: num_filters, num_units, dropout, optimizer. The y-axis shows the accuracy of the neural network. The graph is a line graph with multiple lines, each representing a different combination of the parameters. In the evaluation of CNN architectures with varied hyperparameters, namely the number of filters, units, dropout rate, and optimizer, the obtained accuracies of 0.96 across different configurations highlight a remarkable consistency. The combinations tested, encompassing variations in the number of filters (16, 64), units (64, 128, 256), and a constant dropout rate of 0.2 with the 'Adam' optimizer, all yield identical accuracies. In contrast, one specific set of hyperparameters comprising 16 filters, 64 units, a dropout rate of 0.1, and utilizing the 'SGD' optimizer yielded a comparatively lower accuracy of 0.75.

In Fig. 4, on the second dataset, the highest accuracies achieved were 0.9, obtained from two distinct configurations. The first configuration utilized 16 filters, 64 units, a dropout rate of 0.2, and employed the 'SGD' optimizer. Meanwhile, the second configuration comprised 32 filters, 64 units, a dropout rate of 0.1, and utilized the Adam optimizer. Both configurations yielded the same highest accuracy of 0.9, showcasing the effectiveness of these particular hyperparameter settings on this dataset. The lowest accuracies observed were both recorded at 0.39, resulting from two separate configurations. The first configuration involved 64 filters, 128 units, a dropout rate of 0.1, and utilized the Adam optimizer. Similarly, the second configuration consisted of 64 filters, 64 units, a dropout rate of 0.2, also employing the Adam optimizer. Both configurations resulted in the same lowest accuracy, indicating that these specific combinations of hyperparameters and optimizer choices might not be effectively capturing the essential patterns or features within this particular dataset.

By comparing the hyperparameter values and their corresponding accuracies, it is possible to identify trends and gain insights into the effect of different configurations on

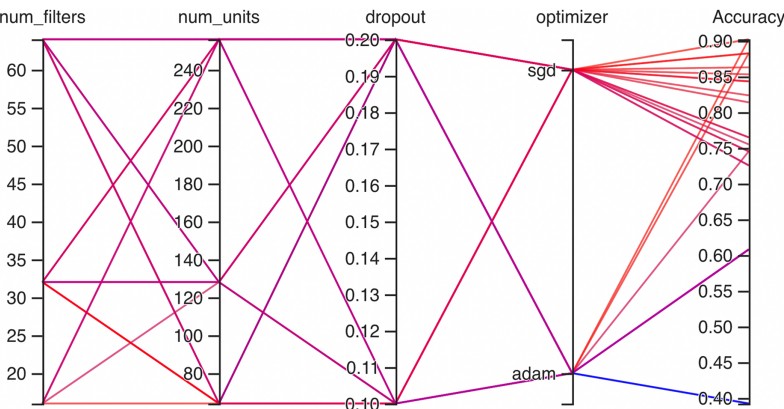

**Figure 4 Working architecture of fine-tuned hyperparametric CNN model for dataset 2.**

the model performance. However, further analysis and experimentation may be required to draw definitive conclusions about the optimal hyperparameter settings, such as cross-validation and statistical significance testing. We did not use Bayesian hyperparameter tuning in our research. Bayesian hyperparameter tuning is a powerful technique, but we decided to use random search for a few reasons. Simplicity and ease of implementation: Random search is a simpler and easier-to-implement technique than Bayesian optimization. This is important because our research aims to provide a practical solution that can be easily adopted by researchers and practitioners who might have limited computational resources.

Baseline comparison: We wanted to establish a baseline comparison against which the performance of our fine-tuned CNN model could be evaluated. By using random search, we can ensure that the improvements achieved by our proposed approach can be attributed to the fine-tuning strategy itself, rather than the specific optimization algorithm.

General applicability: Random search is a versatile method that can work effectively across a wide range of problem domains. We wanted to demonstrate the effectiveness of our fine-tuned CNN approach in a generalizable manner, showcasing its potential applicability to various medical image analysis tasks beyond brain tumor diagnosis. Efficiency and exploration: Random search provides a good balance between exploration and exploitation of the hyperparameter space. While Bayesian optimization is highly efficient in exploitation, random search's exploration-centric nature allowed us to comprehensively explore the hyperparameter configurations, potentially uncovering valuable insights.

## RESULTS

This section explains the experimental results for the hyperparameters of the fine-tuned CNN, explained with various evaluation criteria. The proposed model and the predicted hyperparametric fine-tuned CNN were implemented in Python on a computer system with a GPU of 6 GB GTX 1060, an 8th generation Core i7, and 16 GB of RAM to calculate the results of a brain tumor. The following evaluation criteria were used:

## Evaluation criterion

Several statistical equations are used to test the proposed model for classifying and detecting brain tumors Eqs. (10)–(13). True positive images are those correctly classified and denoted as $Tp$, while true negative images are those incorrectly classified as negative and denoted as $Tn$. $Fp$ stands for the number of incorrectly positive classified images, while $Fn$ stands for incorrectly negative classified images. The statistical metrics accuracy, precision, recall, and F1-score were used to gauge the model output. The F1-score was used to evaluate the outcomes when there was a conflict between accuracy and sensitivity, informative and technical.

$$Precision\ (Pre) = \frac{Tp}{Tp + Fp} \tag{10}$$

$$Recall\ (R) = \frac{Tp}{Tp + Fn} \tag{11}$$

$$F1 - score\ (F1) = \frac{2(Se \times Pre)}{Se + Pre} \tag{12}$$

$$Accuracy\ (Acc) = \frac{Tp + Tn}{Tn + Tn + Fp + Fn} \tag{13}$$

## Model results

The confusion matrix of the test data for the four-class (dataset 1) and two-class (dataset 2) classification is shown in Figs. 5 and 6. The test dataset 1 includes four categories: meningioma, pituitary, no tumor, and glioma while dataset 2 contains yes and no class. The confusion matrix is a square matrix with dimensions equal to the number of classes. In this scenario, with four classes, the confusion matrix is a $n \times n$ matrix; where $n$ is the number of classes. Each cell in the matrix represents a combination of predicted and actual class labels. The numbers within the matrix represent the total number of images utilized for classification. Each entry in the matrix corresponds to the count of images that belong to a specific actual class (represented by rows) and were predicted to belong to a specific predicted class (represented by columns).

The model's ability to generalize new data is measured by the validation and training accuracy, as shown in Figs. 7 and 8, which is determined using a different dataset not used during training. The x-axis represents the number of epochs, and the y-axis represents the accuracy. The graph has two lines, one for training accuracy and one for validation accuracy. The training accuracy line is a dashed blue line, and the validation accuracy line is a dotted orange line. The training accuracy starts at around 0.75 and increases steadily to around 0.98. The validation accuracy starts at around 0.8 and increases steadily to around 0.95. It helps identify overfitting and evaluates the model performance on real-world data. A model may have overfitted to the training data and not generalized well if training accuracy is high, but validation accuracy is noticeably lower. Validation Loss denotes the inconsistency between the predictions of the model and the actual targets in the validation

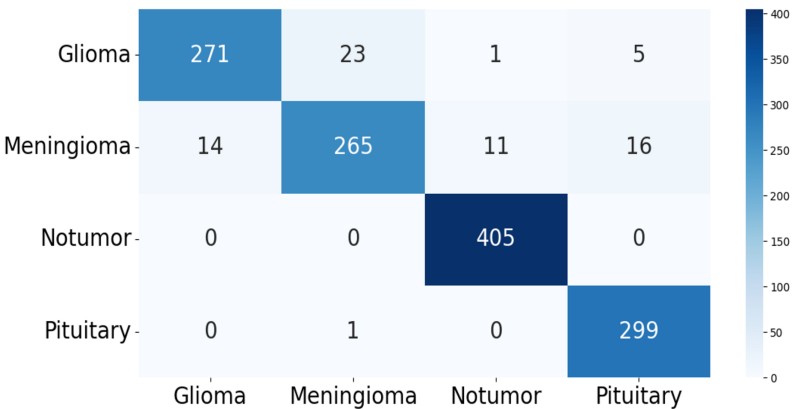

**Figure 5** The confusion matrix generated by the proposed model on the testing data from dataset 1.

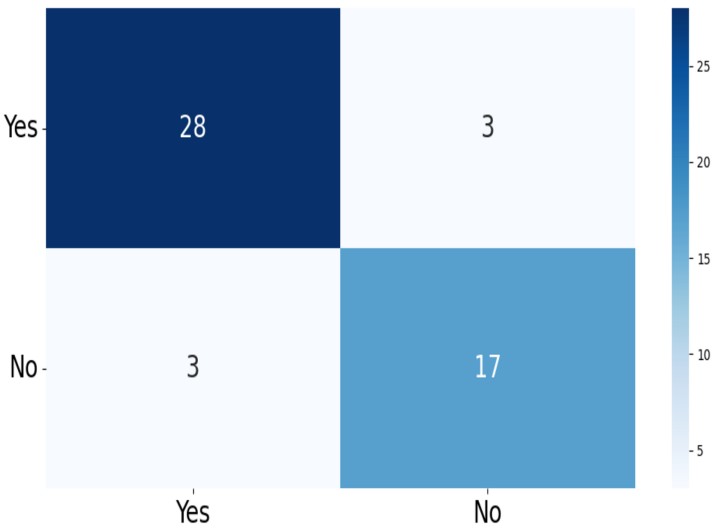

**Figure 6** The confusion matrix generated by the proposed model on the testing data from dataset 2.

dataset, as shown in Figs. 9 and 10 for dataset 1 and dataset 2. It acts as a gauge of the model's effectiveness using previously unobserved data. It is estimated using a loss function, like training loss, and the objective is to minimize it to improve the model's accuracy on the novel, untried samples.

The ROC curve explains the concession among the true positive (sensitivity) and the false favorable rates (specificity minus sensitivity) for various categorization thresholds, as shown in Figs. 11 and 12 for dataset 1 and dataset 2. A point on the curve shows the model's performance at each threshold, which represents a different threshold setting. As the threshold changes, the actual positive rate is drawn against the false positive rate to form the curve. The AUC value calculates the total effectiveness of the model. The
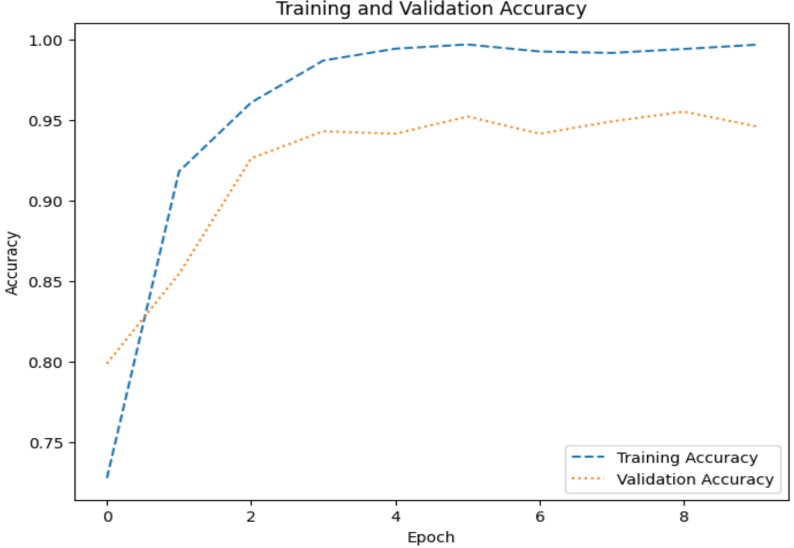

**Figure 7  Accuracy graph for dataset 1.**           

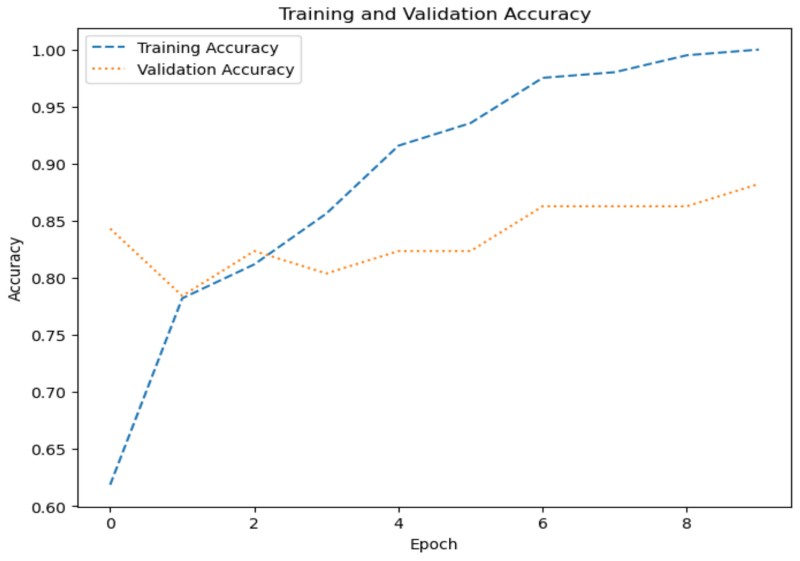

**Figure 8  Accuracy graph for dataset 2.**           

likelihood that a casually designated positive sample will be graded higher than a negative sample is represented by this parameter. The range of AUC values is 0 and 1, with higher values representing improved performance and discrimination capacity.

The statistical information in Tables 4 and 5 summarizes brain tumor classification, and detection performance. Impressive results are displayed in the table, including an average precision, recall, and F1-score of 0.94, showing remarkable accuracy in detecting brain tumors. The accuracy of 0.96 shows the model's overall performance in accurately classifying cases of brain tumors. At the same time, the AUC value of 0.99 indicates outstanding discrimination abilities due to the fine-tuning of hyperparameters of CNN for dataset 1.

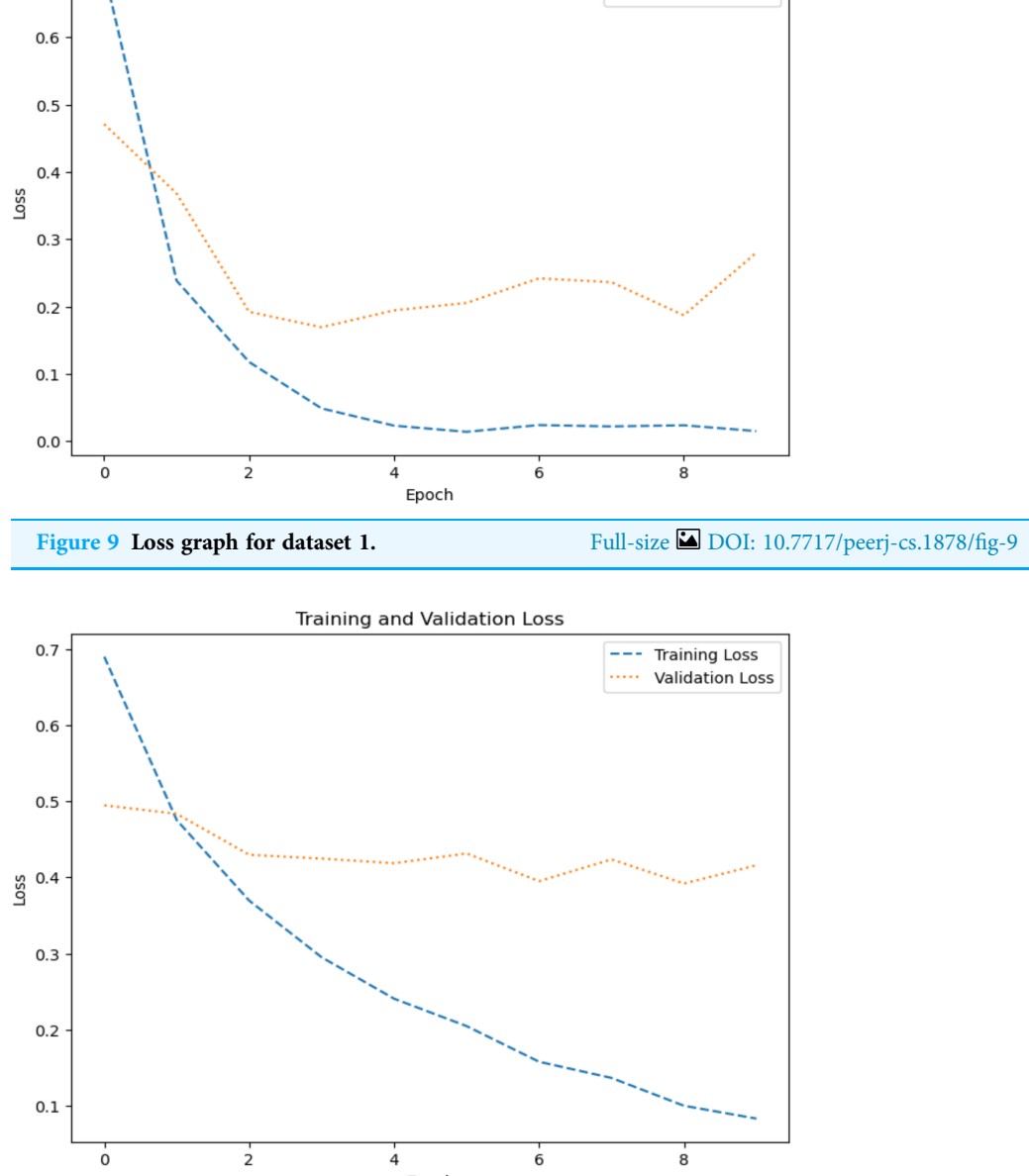

**Figure 9  Loss graph for dataset 1.**  

**Figure 10  Loss graph for dataset 2.**  

In dataset 2, for the 'Yes' class, the precision, recall, and F1 score are all 0.90, indicating a consistent performance in predicting this class. The Support for 'Yes' is 31, implying that this class appeared 31 times in the dataset. The AUC for 'Yes' is also 0.90, indicating good discrimination ability for this class.

The 'No' class shows slightly lower precision, recall, and F1 score at 0.85 but has an AUC of 0.10, which seems unusually low. Typically, AUC values are between 0.5 and 1. This is a point of interest, possibly indicating issues with the model's ability to distinguish the 'No' class correctly. The 'Average' row displays the mean values of precision, recall, and

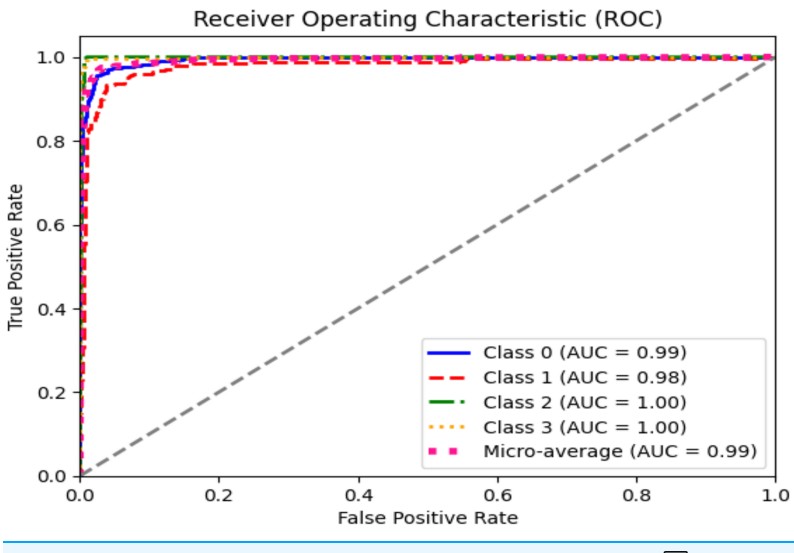

**Figure 11 ROC graph for dataset 1.**

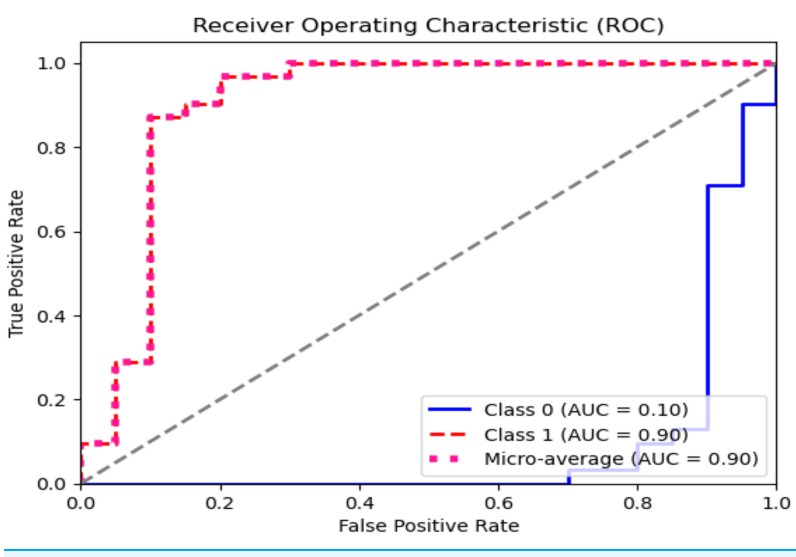

**Figure 12 ROC graph for dataset 2.**

F1 score for both classes, which are all 87.5. The average AUC is 0.5, suggesting good overall discrimination ability across both classes.

Figure 13 show the accuracy values comparison for different hyperparameters of the CNN model. The accuracy values occurred against several filters (num_filters), number of units (num_units), dropout rate (dropout), and optimizer. Two optimizers are used: Stochastic Gradient Descent (SGD) and Adaptive Moment Estimation (Adam). Adam is an adaptive optimizer that dynamically adjusts the learning rate, while SGD uses a fixed one. The highest accuracy value of 96% shows an occurrence rate of 11.1%, and the lowest accuracy value is 75, with an occurrence rate of 2.8%. Generally, the accuracy occurrence range a maximum value of 19.4% for both 92% and 95%.

**Table 4 Statistical results of the model on dataset 1.**

| Class name | Pre | R | F1 | Support | AUC | Accuracy |
|---|---|---|---|---|---|---|
| Glioma | 0.95 | 0.90 | 0.93 | 300 | 0.99 | 0.96 |
| Meningioma | 0.92 | 0.87 | 0.89 | 306 | 0.98 | |
| No tumor | 0.97 | 1.00 | 0.99 | 405 | 1.00 | |
| Pituitary | 0.93 | 1.00 | 0.96 | 300 | 1.00 | |
| Average | 0.94 | 0.94 | 0.94 | – | 0.99 | |

**Table 5 Statistical results of the model on dataset 2.**

| Class name | Pre | R | F1 | Support | AUC | Accuracy |
|---|---|---|---|---|---|---|
| Yes | 0.90 | 0.90 | 0.90 | 31 | 0.90 | 0.88 |
| No | 0.85 | 0.85 | 0.85 | 20 | 0.10 | |
| Average | 87.5 | 87.5 | 87.5 | – | 0.5 | |

## Dataset 1 Accuracy

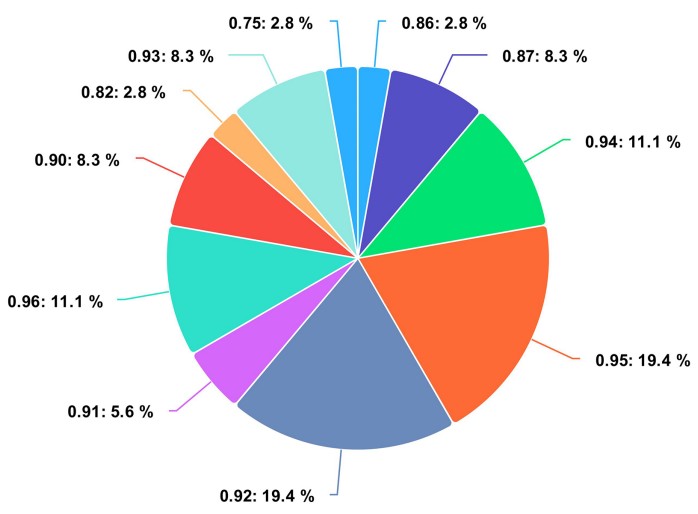

**Figure 13 Comparing the accuracy percentage proportions across various hyperparameter configurations of the CNN model on dataset 1.**

For dataset 2, the accuracy values range from 0.39 to 0.90, indicating the correctness of our proposed model as shown in Fig. 14. The associated percentages vary, showcasing the distribution of how frequently each accuracy level occurs within the dataset. For instance, an accuracy of 0.61 is associated with a high percentage of 27.6%, suggesting a relatively low precision in that specific case. Conversely, some values, like 0.81, 0.82, 0.85, and 0.86, exhibit consistent accuracy percentages of 3.4%, implying more stable performance. Intriguingly, accuracies of 0.39, 0.73, 0.76, 0.84 and 0.90 have percentages of 6.9%, potentially indicating areas of interest or significance within the context of the data.

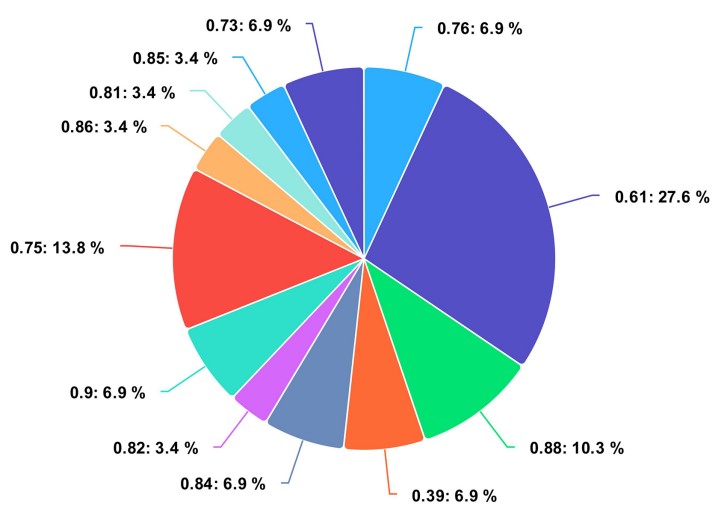

**Dataset 2 Accuracy**

**Figure 14 Comparing the accuracy percentage proportions across various hyperparameter configurations of the CNN model on dataset 2.**

**Table 6 Statistical results compared with state-of-the-art techniques.**

| Reference | Method | Accuracy |
|---|---|---|
| *Srinivas et al. (2022)* | Fine-tuned ResNet-50 with CNN | 0.95 |
| *Ayadi et al. (2022)* | Hybrid approach | 0.90 |
| *Asiri et al. (2023)* | U-Net with fine-tuned ResNet-50 | 0.94 |
| *Bhatele & Bhadauria (2022)* | Hybrid Ensemble | 0.95 |
| *Murthy, Koteswararao & Babu (2022)* | CNN Ensemble | 0.95 |
| Proposed model (dataset 1) | HPCNN | 0.96 |
| Proposed model (dataset 2) | HPCNN | 0.88 |

Moreover, there are instances where relatively high accuracy levels, such as 0.88, are accompanied by a 10.3% percentage, possibly suggesting that precision is balanced with occurrences.

Table 6 presents a comprehensive comparison of various brain tumor classification methods, revealing their respective accuracy scores. Each method's accuracy value indicates the percentage of correctly classified brain tumor images in the dataset. Notably, the "Fine-tuned ResNet-50 with CNN" achieves an accuracy of 0.95, leveraging the combined strengths of the fine-tuned ResNet-50 model and a CNN architecture. Similarly, the "Hybrid approach" attains an accuracy of 0.90 by employing a combination of traditional machine learning algorithms and deep learning models, enhancing classification robustness. The "U-Net with fine-tuned ResNet50" method achieves an accuracy of 0.94, benefiting from U-Net's segmentation capabilities in conjunction with a fine-tuned ResNet-50 model. The "Hybrid Ensemble" and "CNN Ensemble" methods both achieve an accuracy of 0.95 through the ensemble of diverse models and techniques,

resulting in enhanced accuracy. The "Proposed model" stands out with the highest accuracy of 0.96, outperforming all other methods, potentially advancing brain tumor diagnosis in medical imaging applications. In one of our experiments, we conducted tests on a smaller dataset comprising 253 images from two distinct classes. Despite the reduced data size, our system demonstrated remarkable performance, achieving an accuracy of 88%. This outcome underscores the robustness and versatility of our hyperparameter tuning approach in scenarios where data availability is constrained.

## CONCLUSION

In this study, we developed a new CNN model for brain tumor diagnosis. We carefully tuned the hyperparameters of the model, including the number of filters, filter size, stride padding, pooling techniques, activation functions, learning rate, batch size, and layer configuration. This resulted in a significant improvement in diagnostic accuracy. We tested our model on two publicly available brain tumor MRI datasets. For the first dataset, which contains 7,023 brain images across four classes, our model achieved an accuracy of 96%, along with an average precision, recall, and F1-score of 94.25%. For the second dataset, which contains 253 images, our model achieved an accuracy of 88%, along with precision, recall, and F1-score values of 87.5%. These results demonstrate the effectiveness of our CNN model for brain tumor detection and classification. Our model outperforms existing methods in terms of accuracy and efficacy, and it offers the potential to improve the precision and efficiency of brain tumor diagnosis. This could lead to earlier detection and treatment of brain tumors, which could save lives. In addition, our study demonstrates the importance of hyperparameter optimization in CNN models for medical diagnostics. By carefully tuning the hyperparameters of our model, we were able to achieve significant improvements in accuracy. This suggests that hyperparameter optimization could be used to improve the performance of CNN models for other medical applications, such as cancer detection and diagnosis.

### Funding

This work is supported by the Deanship of Scientific Research, Najran University, Kingdom of Saudi Arabia under the Distinguished Research funding program grant code number (NU/DRP/MRC/12/28). The funders had no role in study design, data collection and analysis, decision to publish, or preparation of the manuscript.

### Grant Disclosures

The following grant information was disclosed by the authors:
Deanship of Scientific Research, Najran University. Kingdom of Saudi Arabia, Distinguished Research funding program: NU/DRP/MRC/12/28.

### Competing Interests

The authors declare that they have no competing interests.

## Author Contributions

- Abdullah A. Asiri conceived and designed the experiments, analyzed the data, prepared figures and/or tables, and approved the final draft.
- Ahmad Shaf conceived and designed the experiments, performed the computation work, prepared figures and/or tables, and approved the final draft.
- Tariq Ali conceived and designed the experiments, analyzed the data, authored or reviewed drafts of the article, and approved the final draft.
- Muhammad Aamir conceived and designed the experiments, performed the experiments, performed the computation work, authored or reviewed drafts of the article, and approved the final draft.
- Muhammad Irfan performed the experiments, analyzed the data, performed the computation work, authored or reviewed drafts of the article, and approved the final draft.
- Saeed Alqahtani performed the experiments, prepared figures and/or tables, and approved the final draft.

## Data Availability

The code is available at GitHub and Zenodo:

- https://github.com/iamshaf/CNN-HyperParameter.

- iamshaf. (2024). iamshaf/CNN-HyperParameter: HPCNN (HPCNN). Zenodo. https://doi.org/10.5281/zenodo.10476557.

Dataset 1 is available at Kaggle: https://www.kaggle.com/datasets/masoudnickparvar/brain-tumor-mri-dataset.

Dataset 2 is available at Kaggle:

https://www.kaggle.com/datasets/navoneel/brain-mri-images-for-brain-tumor-detection.

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
