# Peer review of "Enhancing brain tumor diagnosis: an optimized CNN hyperparameter model for improved accuracy and reliability"

_PeerJ Computer Science, doi:10.7717/peerj-cs.1878_

## Round 0.1 · original submission · Major Revisions

Dear Authors,

Your paper has been reviewed. Based on the reviewers' opinions, the manuscript needs major revisions before being accepted for publication in PeerJ Computer Science. More precisely, the main issues are related to the unclear presentation and the necessity to collocate better your contribution concerning publications on the same subject from 2021-2023.

Thank you for your interest in PeerJ Computer Science.

Reviewer 1 ·

Basic reporting

Reviewer’s Report on the manuscript entitled:

Enhancing brain tumor diagnosis: an optimized CNN hyperparameter model for improved accuracy and reliability

The authors utilized CNN by tuning its hyperparameters. They compared it with other models and successfully applied it to publicly available brain tumor MRI datasets. The manuscript is generally well-written and interesting. I have some suggestions for improvement:

Line 97. Please rewrite this sentence in a better way.

Lines 122-125. This is a redundant paragraph. You can elaborate more on how you organized the manuscript and what models, datasets, used.

Line 127. You can remove “Authors” and simply write “Deng et al., (2009) implemented….”
Line 148. Again, please replace “In another paper (Xie et al., 2022)…” with “Xie et al., (2022)….”.
Similarly, for other places as appropriate.

Figures 5 to 14. Please enlarge the font size of all the texts and numbers in these figures and ensure their resolution is at least 300 dpi. (high quality).

Figures 13 and 14. These figures need explanation. In their captions, please write what those percentages for hyperparameters mean?

Tables 1 and 2 and 7. Please add another column that says what dataset was used in each paper, so the reader can see the comparison.

Table 1. Row 2. Chen et al. (2019b). This is an application to forest fire not MRI. I suggest only limit Tables 1 and 2 to MRI images for tumor detection.


Line 228. Please make the format of these variables the same as in the equations (i.e., math, italic). Please do the same for other equations.

Line 303. Please define AUC and ROC. All the abbreviations should be defined the first time they appear. Please also provide some references for them.

Line 408. Do you mean Table 7?

Line 410. Did you use techniques, such as gradient clipping and early stopping to speed up computation and prevent over-fitting issues in your model? Please discuss in the light of the following article by Naushad et al. (2021) where they described these in detail showing that ResNet-50 achieved an accuracy of 99.17% after fine tuning its parameters and applying techniques, such as early stopping:
https://doi.org/10.3390/s21238083
You can also include Chen et al. (2019b) when you describe the over-fitting issue.


Thank you

Regards,

Experimental design

Please add a flowchart figure showing the input, output, and method used. This way a general reader can easily and quickly see what you have done in this paper.

Validity of the findings

No comment

Additional comments

Please carefully proofread the manuscript and correct grammar/style/typo issues. Please also improve the quality of the figures.

Reviewer 2 ·

Basic reporting

This paper proposes a hyperparametric model for optimizing CNN networks to enhance the accuracy and reliability of brain tumor diagnosis, providing some new insights into optimizing network efficiency. However, the paper suffers from the problem of unclear presentation, and the authors are advised to revise the paper carefully to improve its quality.

Experimental design

The main focus of the article is on optimizing hyperparameters to improve accuracy as well as reliability. Focusing on describing and emphasizing the objective indicators, a large number of comparisons were accomplished in terms of data analysis, but subjective comparison results were lacking, and it is suggested to increase the visualization results in the process of tumor detection and classification, as well as the comparison results with other methods. It is suggested to select several classical and the latest methods for hyperparameter optimization experiments to verify the effectiveness of the methods and make the paper more comprehensive.

Validity of the findings

In the comparison of the literature table 1 of the existing work, the latest reference is in 2020, and it is recommended that the literature research of the last three years should be added appropriately to make the comparison results more persuasive.

Additional comments

1.The first three paragraphs of the introduction section are overly redundant in describing brain tumors as a medical condition. It is recommended that the introduction section be rewritten to highlight the focus of the article's research.
2.The resolution of the figures is insufficient. It is recommended that vector graphics be used in the paper to increase the clarity of the images.
3.In line 304, Figure 2 should be Figure 3, and in line 388, Figure 7 should be Figure 13.
4.The explanations of the diagrams are inconsistent with what the pictures reflect, and it is recommended that the references to the diagrams be checked throughout for correctness and standardization.
5.Please indicate the meaning of the different colored lines in Figures III and IV.
6.Figures 7, 8, 9, and 10 are the same type, but figures 8 and 10 have the caption inside the figure, while figures 7 and 9 do not. It is recommended that the four figures be in a uniform form.
7.The font size in Figure 2 is too large and it is recommended to match the font size of the figure notes.
8.Suggests a detailed explanation of the formulas in the article. and please check throughout the formula for standardization and consistency of capitalization, for example, Polling Size is capitalized in equation 9.
9.In Figures 7 and 8, blue dashed lines and orange dashed lines are used to represent training accuracy and validation accuracy respectively. But in Figures 9 and 10, the author is trying to illustrate the loss curve, and still using blue and orange is not appropriate, so it is suggested that the loss curve illustration be changed to another color.

·

Basic reporting

The paper is well-written and well-organized.
In Table 1, all the years of the references are 2018, 2019, and 2020.
I suggest adding more recent papers( 2022 and 2023).

Experimental design

- Add in Table 1, the advantages of each cited paper
-Add a comparison with the new models that used pruning weight.

Validity of the findings

no comment

Additional comments

No comment

---

## Round 0.2 · accepted · Accept

Dear Authors,

Since you have addressed all of the reviewers' comments, your paper can be accepted for publication.

Reviewer 2 ·

Basic reporting

The authors have revised the paper according to the reviewer's comments.

Experimental design

no comment

Validity of the findings

no comment

Additional comments

no comment